# Learning abstract structure for drawing by efficient motor program induction

**Lucas Y. Tian**[1,2,3], **Kevin Ellis**[1,2], **Marta Kryven**[1,2], and **Joshua B. Tenenbaum**[1,2]

[1]Brain and Cognitive Sciences, MIT
[2]Center for Brains, Minds and Machines
[3]Laboratory of Neural Systems, Rockefeller University
`{lyt,ellisk,mkryven,jbt}@mit.edu`

## Abstract

Humans flexibly solve new problems that differ from those previously practiced. This ability to flexibly generalize is supported by learned concepts that represent useful structure common across different problems. Here we develop a naturalistic drawing task to study how humans rapidly acquire structured prior knowledge. The task requires drawing visual figures that share underlying structure, based on a set of composable geometric rules and simple objects. We show that people spontaneously learn abstract drawing procedures that support generalization, and propose a model of how learners can discover these reusable drawing procedures. Trained in the same setting as humans, and constrained to produce efficient motor actions, this model discovers new drawing program subroutines that generalize to test figures and resemble learned features of human behavior. These results suggest that two principles guiding motor program induction in the model - abstraction (programs can reflect high-level structure that ignores figure-specific details) and compositionality (new programs are discovered by recombining previously learned programs) - are key for explaining how humans learn structured internal representations that guide flexible reasoning and learning.

## 1   Introduction

A long-term goal of Artificial Intelligence (AI) is to build machines that can quickly learn to solve new problems. Inspiration may be gained from studying human intelligence. People readily learn to perform many kinds of tasks without extensive supervision or direct experience, such as inferring what kinds of objects a new word is referring to, or rapidly learning new video games [1]. These abilities are partly supported by learned internal models, or inductive biases, whose structure represents regularities useful for reasoning about new situations (e.g., the hierarchical structure of object categories, or rules common across different video games) [2, 3, 4, 5, 6, 7]. In this work we adopt a scientific, rather than engineering, goal: to probe diagnostic elements of how humans *acquire* structured prior knowledge, and to understand it in computational terms.

To study such learning in a controlled setting, we introduce a drawing task to investigate rapid, few-shot updating of structured internal models. On its surface, the task is simple: to copy, by drawing on a touchscreen, a series of novel visual figures. However, *how* people draw provides rich insight into their internal representations and prior knowledge. This is intuitive when one considers the inherent ambiguity in how even simple line-drawings should be copied. How do different line segments group into coherent objects? How are different objects related? Drawing has therefore been studied for insight into how reasoning and problem-solving is guided by structured prior knowledge, including concepts as diverse as geometry, real-world objects, and geological formations [8, 9, 10, 4, 11, 12, 13].

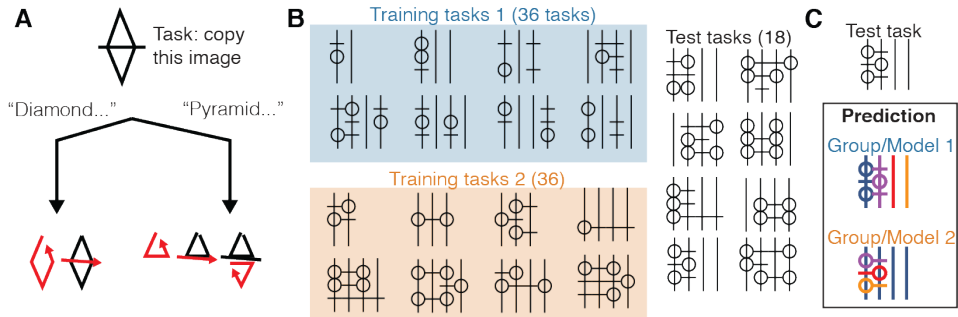

Figure 1: Background and overview. **(A)** Prior knowledge influences how people draw. Consider how one might copy the drawing on top. Copying behavior tends to differ depending on what structural description is associated with the object - "Diamond with a cross-line" vs. "Pyramid and its reflection" [8]. **(B)** Representative tasks in Training sets 1 and 2, and in the common Test set. **(C)** Hypothesized behavior on test tasks that would be diagnostic of learned abstract structure.

For example, consider copying the figure in Figure 1A. The order of strokes people use to copy this figure depends whether it is described as a "diamond with a cross-line" or an "Egyptian pyramid and its reflection on water" (Figure 1A) [8].

How the prior knowledge that guides drawing is learned has been the subject of numerous descriptions of drawing behavior across children and adults (e.g., [13, 8, 9, 14]). Other studies have described single-session learning of specific objects, such as cubes or prisms (e.g., [15, 16]). Our goal differs from this prior work in combining a focus on learning that is rapid and generalizable with a formal computational account of this rapid learning.

Motivated by prior empirical and theoretical studies (and supported *post hoc* by behavior in this study), we model learning by incorporating two key principles: *abstraction* and *compositionality*. Abstract refers to higher-order structure that is independent of figure-specific features. In principle, this supports reasoning that is flexible even in novel situations that differ in lower-level features; e.g., the concept of *repeat* can apply to any simple drawn figure [7, 17, 3, 18, 19]. Compositional refers to complex concepts learned by combining simpler conceptual building blocks. E.g., *repeat* and *hexagon* can be combined to draw an object along the perimeter of an imagined hexagon. Compositionality enables complexity and variation that extrapolates beyond direct training experience [5, 20, 7, 21, 17]. Our model realizes abstraction and compositionality through *program induction*. Concretely, it attempts to copy a given figure by synthesizing an abstract, compositional graphics program that, when rendered, produces the figure. The model learns by building an increasingly complex "library" of drawing code from experience on the same training figures given to human subjects, using recent neuro-symbolic program induction algorithms [22, 23].

Our behavioral data provides evidence that humans indeed perform few-shot updating of their inductive biases by learning program-like drawing procedures that guide generalization behavior. We describe a program-induction algorithm that discovers new abstract, compositional, drawing routines given the same limited training data given to humans. Moreover, the model's learned drawing behavior captures certain diagnostic features of how humans generalize. These results suggest that abstraction and compositionality are key principles for explaining how humans rapidly learn program-like structure that guides reasoning and planning in drawing.

## 2   A neuro-symbolic model for learning compositional drawing programs

Our model is inspired by prior approaches modeling handwriting and drawing [24, 25, 10], AI models that learn structured inductive biases in other cognitive domains [26, 27, 17, 28], and models of perception as inference of symbolic descriptions [29, 19, 30, 19, 31, 32]. We model drawing behavior based on programs, or symbolic procedures representing a description of a drawing's parts - here, simple primitives such as lines and circles - and higher-order relations - e.g., repetition and hierarchy. For a given image, the model infers how to draw it by performing probabilistic inference over a space of drawing programs. Learning works by estimating a prior over programs in a hierarchical Bayesian fashion. Estimating this prior involves inducing new drawing subroutines that are useful for multiple drawing tasks, effectively caching and reusing motor program schemas. These learned

subroutines are *abstract* and *compositional*. Finally, for comparison with human behavior, we convert these programs into low-level action trajectories. This conversion rests on a third principle relevant for action planning -*motor efficiency* - which is formalized by modeling a proxy for motor costs.

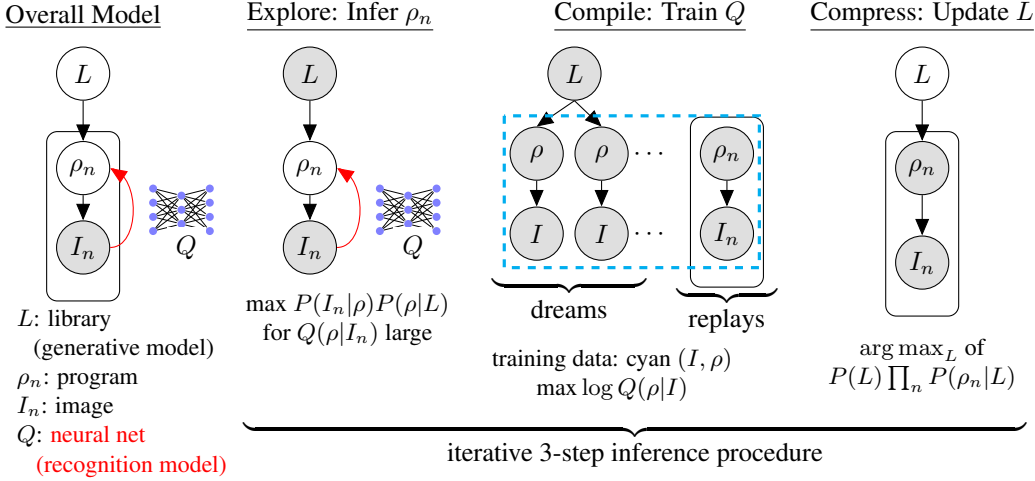

Figure 2: The Bayesian neurosymbolic program induction algorithm that underlies our computational model, based on [23, 33]. Left ("overall model"): Each observed image $I_n$ is explained using a latent program $\rho_n$. The prior or inductive bias is modeled by an inventory or "library" of learned primitives, $L$. A neural network recognition model (red arrows) learns to map from images to a distribution over source code of programs likely to explain that image. Conditional distribution output by the network is notated $Q(\cdot|\cdot)$. Inference iterates through **Explore**, which searches programs ordered under $Q$ and rescores them under true posterior $P(\cdot|L, I_n)$; **Compile**, which trains the neural network $Q$ to search for image-explaining programs, training both on replays of programs from Explore and "dreams," or samples from the learned prior; and **Compress**, which updates the prior by compressing out new compositional code abstractions which are incorporated into the library $L$.

**Program-Induction Model**  We treat drawing as Bayesian inference over the most likely program $\rho$ that generated each image. Programs are sampled from a generative model defined by a library of primitives $L$, using a neurally-guided systematic search. Models are initialized at training onset with a library of simple drawing- and geometry-related primitives, which are plausibly accessible to humans even before they start the study (Table 1). On each "trial", the model is given an image $I$ and returns a $\rho$ maximizing:

$$P(\rho|I, L) \propto \underbrace{P(I|\rho)}_{\text{likelihood: } \mathbb{1}[\rho \text{ draws } I]} \times \underbrace{P(\rho|L)}_{\text{description-length prior}} \tag{1}$$

where $\mathbb{1}[\rho \text{ draws } I]$ is 1 if the rendered program copies the image, and 0 otherwise. For test images we allow "partial credit" by relaxing the likelihood function to pixel-wise L2 distance (see Suppl. Sect. 1.2).

The model learns from experience drawing. Given training images $\{I_n\}_{n=1}^{N}$ the model updates its library $L$ by searching to maximize:

$$P(L|\{I_n\}_{n=1}^{N}) \propto \underbrace{P(L)}_{\text{description-length prior}} \times \prod_{n=1}^{N} \sum_{\rho} P(I_n|\rho)P(\rho|L) \tag{2}$$

Equations 1 and 2 are intractable, because they require computing the infinite set of all possible programs. We approximate inference using an iterative approach based on the DreamCoder program synthesis algorithm (Figure 2, Suppl. Sect 1; see [23, 33]). This alternates between inferring a program for each image (**Explore**), updating the library $L$ with discovered subroutines used in program solutions across images (**Compress**), and training a neural network, $Q(\rho|I)$, to predict a

Table 1: Starting primitives in library $L$. The model learns an inductive bias over programs by inducing new subroutines that are built by recombining these primitives.

| Primitive | Arg. types[1] | Returns | Description |
|---|---|---|---|
| line | none | $D$ | Line with endpoints at (0,0) and (1,0) |
| circle | none | $D$ | Unit circle centered at (0,0) |
| repeat | $(D, n, T)$ | $D$ | Drawing accumulated by transforming $D$ by $T$ $n$ times |
| transform | $(D, T)$ | $D$ | Applies affine transformation $T$ |
| reflect | $(D, \theta)$ | $D$ | Reflects across axis defined by $\theta$ |
| connect | $(D, D)$ | $D$ | Union of two drawings |
| affine | $(s, \theta, d, d, o)$ | $T$ | Scaling ($s$), rotation ($\theta$), translation ($d$, $d$) in order $o$ |

probability distribution over programs $\rho$ likely to explain image $I$ (**Compile**). Learned subroutines can be abstract (e.g., taking as input arbitrary parameters or subprograms).

**Converting programs to action trajectories**  Programs $\rho$ are structural descriptions that do not represent ordering of strokes. For example, a program that translates a vertical line four times could correspond to either a left-to-right or a right-to-left drawing sequence. Thus, we "ground" these programs into possible action trajectories $t$, defined for model and humans as an ordered list of segmented "strokes", each stroke summarized by a feature vector (see "Analysis of behavior"). Each program $\rho$ generates a set of *admissible* $t$, which includes all trajectories whose stroke-sequence can be aligned with the program's syntax tree.

The probability of $t$ given program $\rho$, assuming no reweighting of trajectories based on motor cost (see below), is therefore given by:

$$P(t|\rho) = \frac{\mathbb{1}\left[t \text{ admissible for } \rho\right]}{\sum_{t'} \mathbb{1}\left[t' \text{ admissible for } \rho\right]} \tag{3}$$

**Reweighting action trajectories by motor efficiency cost**  Drawing is influenced by a variety of motor constraints [13, 10, 8, 24]. Many of these constraints, such as those due to biomechanics, are not related to the model's drawing primitives, but are crucial for enabling more human-like behavior in the model. To approximate these constraints, we assign to each action trajectory a score summarizing "efficiency", based on motor-level statistics of human behavior.

All possible action trajectory permutations $t$ for a given program $p$ are assigned an efficiency cost. We define a *feature extractor* $\phi(\cdot)$ that maps a trajectory $t$ to a real-valued feature vector $\phi(t)$ (note that this is different from the stroke-level feature vector previously described) with four elements based *a priori* on previously described motor-level biases in drawing [24, 8]: $start$ (position of first stroke relative to top-left corner), $distance$ (total movement distance), $direction$ (direction of movements relative to the diagonal), and $verticality$ (bias for vertical transitions) (see details in Suppl. Sect 2.3). Given an input image $I$, the model predicts a drawing trajectory $t$ with probability

$$P(t|I) = \mathbb{1}\left[t \text{ draws } I\right] \frac{\exp\left(-\theta \cdot \phi(t)\right)}{\sum_{t'} \mathbb{1}\left[t' \text{ draws } I\right] \exp\left(-\theta \cdot \phi(t')\right)} \tag{4}$$

where $\mathbb{1}[t \text{ draws } I]$ is 1 when $t$ renders into $I$, and 0 otherwise. $\theta$ is a weight vector, so that $\theta \cdot \phi(t)$ is the cost of trajectory $t$.

Given a set of $N$ training images $\{I_n\}_{n=1}^N$ and paired action trajectories $\{t_n^s\}_{n=1}^N$ for subject $s$, the model estimates for each subject $\theta^s$ via regularized maximum likelihood,

$$\theta^s = \arg\min_{\theta^s} \sum_{n=1}^N \underbrace{-\log P(t_n^s|I_n^s)}_{\text{depends on } \theta^s;\, \text{Eq. 4}} + \lambda||\theta^s||_2^2 \tag{5}$$

with a suitable coefficient of regularization $\lambda$. Finally, action trajectories were reweighted as described below in "Models".

**Models**   The full model (Hybrid) combined program induction with motor efficiency costs. It was compared to two "lesioned" models testing the role of program induction (Motor Cost), motor efficiency costs (Program Induction), and a Null model lacking any learned components (Table 2).

Table 2: Models used in this study

| Model | Training set | Abbrev. | Description |
|-------|-------------|---------|-------------|
| Null | none | Null | All trajectories, equal probability |
| Motor Cost | 1, 2 (motor data) | MC1, MC2 | All traj., w/group-specific cost |
| Program Induction | 1, 2 (images) | PI1, PI2 | Admissible traj., all equal probability |
| Hybrid | 1, 2 (mot. & im.) | HM1, HM2 | Admissible traj., w/across-group cost |

Given an input image, models returned trajectories with weights represented as probabilities:

$$P_{Null}(t|I) = \frac{\mathbb{1}\left[t \text{ admissible for } I\right]}{\sum_{t'} \mathbb{1}\left[t' \text{ admissible for } I\right]} \tag{6}$$

$$P_{MC}(t|I, \theta^{group}) = \frac{\mathbb{1}\left[t \text{ admissible for } I\right] \exp(-\theta^{group} \cdot \phi(t))}{\sum_{t'} \mathbb{1}\left[t' \text{ admissible for } I\right] \exp(-\theta^{group} \cdot \phi(t'))} \tag{7}$$

$$P_{PI}(t|\rho^{best}) = \frac{\mathbb{1}\left[t \text{ admissible for } \rho^{best}\right]}{\sum_{t'} \mathbb{1}\left[t' \text{ admissible for } \rho^{best}\right]} \tag{8}$$

$$P_{HM}(t|\rho^{best}, \theta^{gen}) = \frac{\mathbb{1}\left[t \text{ admissible for } \rho^{best}\right] \exp(-\theta^{gen} \cdot \phi(t))}{\sum_{t'} \mathbb{1}\left[t' \text{ admissible for } \rho^{best}\right] \exp(-\theta^{gen} \cdot \phi(t'))} \tag{9}$$

Where $\theta^{gen} = \frac{1}{S}\sum_{s=1}^{S}\theta^s$ is single subject-wide average vector, $\theta^{group}$ is a group-specific vector averaged across all subjects that were trained on the same set of tasks as the model (1 or 2), and $\rho^{best}$ is the highest scoring program given $I$ (see Suppl. Sect 1.2). Using $\theta^{gen}$ for the Hybrid model ensures that training-related effects are due to learning in the program induction model.

**Scoring model-human distance**   For each combination of test image $I$, human $h$, and model $m$, we measured the distance between human behavior and model predictions by:

$$d(h, m, I) = \sum_{t'} dist_{traj}(t', t^h) P_m(t'|I) \tag{10}$$

$P(t')$ is defined for each model as in "Models". $dist_{traj}(t, t')$ computes a scalar distance between trajectories $t$ and $t'$, based on a string-edit distance (see Suppl. Sect. 2.2).

## 3   Experiments

### 3.1   Methods

**The drawing task**   Humans and models copy visually-presented figures defined by simple objects and composed geometric rules (Figure 1). Humans were randomly split into two groups whose training sets differed in higher-order structure. Generalization was tested on a single common set of tasks. Therefore, differences in behavior on test stimuli are attributable to learning on the training sets.

**Subjects**   Subjects [N = 104 (58 M, 44 F, 2 excluded due to excessive behavioral errors or errors in saving data), Age = 35.0 +/- 9.3 (mean/SD)] were recruited on Amazon Mechanical Turk and paid $3.00 for 15-20 minutes. Subjects gave informed consent. The study was approved by the MIT Institutional Review Board.

**Stimuli**   See examples in Figure 1C. Two sets of stimuli (Tasks 1 and 2) were generated using different probabilistic algorithms. Both had repeated vertical lines (2, 3, or 4), but differed in the other components. Set 1 had vertically grouped strokes (lines and circles) superimposed on the vertical lines, while Set 2 had horizontally-oriented groups of strokes, sampled from an library of objects (e.g. dumbbells (o–o), lollipops (–o) and poles (—)). We randomly generated 250 stimuli for each training set, from which we manually selected 36 representative samples. The common Test set included 18 manually designed ambiguous images.

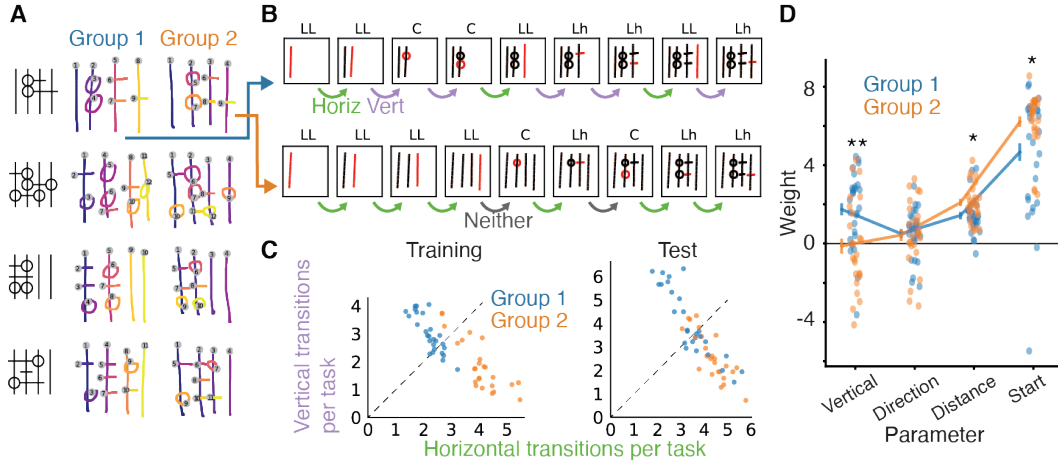

Figure 3: Single-session learning of structure in drawings. **(A)** Example drawing trajectories for two subjects (columns) on four test tasks (rows). Stroke order is indicated by both color (purple to yellow) and the number in grey dot. Grey dots also indicate start positions. **(B)** Example segmentations of behavior into action trajectories. Letter codes indicate stroke categories. **(C)** Stroke transitions were directionally biased for both Training and Test tasks. **(D)** MC feature weights for behavior on Test stimuli. Positive corresponds to a bias for transitions that are vertical, towards bottom-right, low distance, and for first stroke at top-left. *, **, p<.05, .005, t-test.

**Procedure**   The experiment was presented in a web browser using PsiTurk [34] and Raphael SketchPad[2] on a touchscreen device (phone, tablet, or laptop). The instructions read: *You will learn to write letters from an alphabet of an alien civilization recently discovered by astronauts. Scientists would like to study how people learn to write new alphabets. Your task is to copy the letters. Try to be quick, but it is also important to be accurate! Letters are taken from the same alphabet. But letters get harder over time, so try to learn from the earlier trials!* On each trial a single figure was presented top-center of the screen. The subjects copied it on a "sketchpad" directly below without time constraints or evaluative feedback. Subjects first saw seven simple figures (e.g the first three figures in Figure 1C), followed by 13 training figures of varying difficulty. Next, subjects copied the 18 test figures intermixed with the remaining 16 training figures, in orders randomized for each subject.

**Analysis of behavior**   Raw motor trajectories were segmented into discrete *strokes*, defined by periods of uninterrupted touch, and each stroke was summarized by a feature vector $\psi_{stroke}$ = ($category$, $startLocation$, $center$, $row$, $column$) (details in Suppl. Sect 2.1). Each action trajectory was thus defined by an ordered list of strokes: $(\psi_{stroke}^1, \psi_{stroke}^2, ...)$.

## 3.2   Human results

**Rapid learning of structure in drawings**   We expected that a behavioral readout of learning would be for subjects trained on Task 2 (horizontally structured objects) to produce a relatively higher frequency of horizontal transitions, compared to subjects trained on Task 1 (vertically structured objects) (Figure 1C). Indeed, we found this to be the case (Figure 3A-C). We confirmed that this difference in vertical vs. horizontal biases remained even after accounting for other potential changes to behavior. We jointly fit parameters describing four different motor features: *Start, Distance, Direction*, and *Verticality*, fit separately for each subject using the Motor Cost model. While we found weak, but significant, differences in *Start* and *Distance* between the two groups, we found a relatively strong difference in *Verticality* weights, consistent with the previous analysis of transition frequencies (Figure 3D). These results show that training led to strong apparent biases for vertical vs. horizontal transitions.

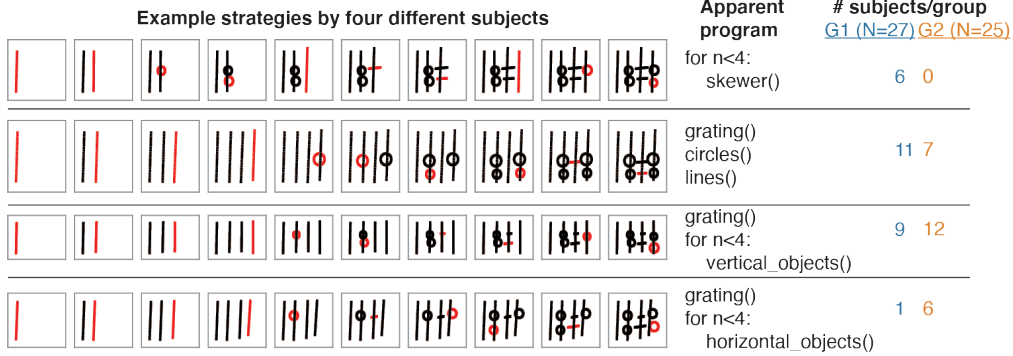

| Example strategies by four different subjects | Apparent program | # subjects/group G1 (N=27) G2 (N=25) |
|---|---|---|
| | for n<4:<br>skewer() | 6   0 |
| | grating()<br>circles()<br>lines() | 11   7 |
| | grating()<br>for n<4:<br>vertical_objects() | 9   12 |
| | grating()<br>for n<4:<br>horizontal_objects() | 1   6 |

Figure 4: Program-like structure in behavior. Four example subjects depicting dominant strategies in our dataset (left), apparent program-like structure (middle), and frequencies of these strategies for the two training groups (see also Suppl. Figure 3).

**Program-like structure in behavior**  Did differences in vertical and horizontal biases reflect changes in lower-level motor preferences, or more abstract biases? Consistent with abstraction, subjects' qualitative behavior appeared to be well-described by programs with abstract structure (Figure 4). In particular, four program-like strategies were prominent across subjects. The "skewers" strategy involved drawing a vertical line, immediately followed by the objects "skewered onto it" (Figure 4, top row). "Skewers" was only observed in Group 1 subjects. Other strategies involved first drawing the vertical gratings followed by different ways of drawing the smaller objects (Figure 4, rows 2-4). We quantitatively assigned one strategy to each subject based on the distribution of Motor Cost model parameters, extended with two additional parameters capturing biases for (1) perseverating on a given category of objects (e.g., circle-circle-circle...), and (2) finishing an entire "skewer" before moving on to the next (see Suppl. Sect 2.4). Group 1 and Group 2 subjects tended to use different program-like strategies (Figure 4, right).

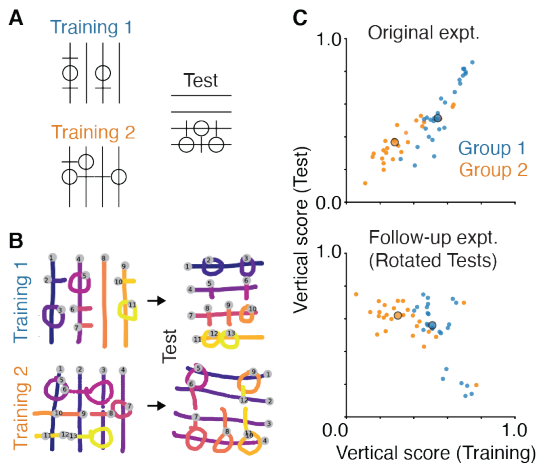

Figure 5: Evidence for abstract generalization in a followup study with rotated test stimuli. **(A)** Test stimuli are rotated relative to the original experiment in Figure 3. **(B)** Example drawings for two subjects showing that they retained program-like biases evident for rotated Test tasks. **(C)** Summary analysis. "Vertical score" is computed as $V/(V + H)$, where $V$ and $H$ are average vertical and horizontal transitions per task. Large dots indicate medians.

**Evidence for abstract generalization**  On a new set of subjects, we performed a modified experiment to further test whether subjects indeed learned abstract program structure. We reasoned that abstract programs should persist if the Test stimuli were rotated (Figure 5A). However, if subjects learn only the horizontal vs. vertical motor biases, we would expect this bias to remain unchanged. We found that directional biases changed orientation when the test stimuli were rotated (Figure 5B,C). In the original (non-rotated) experiment subjects in Group 1 exhibited a stronger vertical bias than those in Group 2 during Training, and this effect carried over to Testing (replotted in Figure 5C). However, in this second modified experiment, while Group 1 subjects still exhibited a stronger vertical bias during training, they preferred horizontal transitions when tested on rotated stimuli (Figure 5C). This flexible adaptation of directional biases in a manner that mimics the orientation of the stimuli is consistent with the learning of abstract programs.

## 3.3 Modeling results

**Program induction** We trained a pair of models on either Training sets 1 (HM1) or 2 (HM2), initialized with a common set of simple primitives (Table 2; some drawn in Figure 6A). The models successfully learned new program subroutines (examples in Figure 6B; entire set in Supplemental Figures 1 and 2). This was reflected in unconditioned samples from learned priors (i.e., "dreams"), which exhibited task-related structure that differed based on training. For example, Model 1 dreams included vertical "skewers", while Model 2 dreams included horizontal "barbells". Some dreams also extrapolated from the training stimuli (Figure 6D). In contrast, untrained models did not exhibit task-related dreams (Figure 6D, "Baseline").

Trained models performed well on the test tasks [mean/SD of 1.9/1.7 (HM1) and 0.35/0.76 (HM2) mistakes (missed or extra strokes) per task (out of 11.6)]; in contrast, the untrained model was unable to solve any test tasks. Moreover, the two trained models often produced different solutions to the same task, coarsely resembling solutions produced by trained humans (Figure 6C).

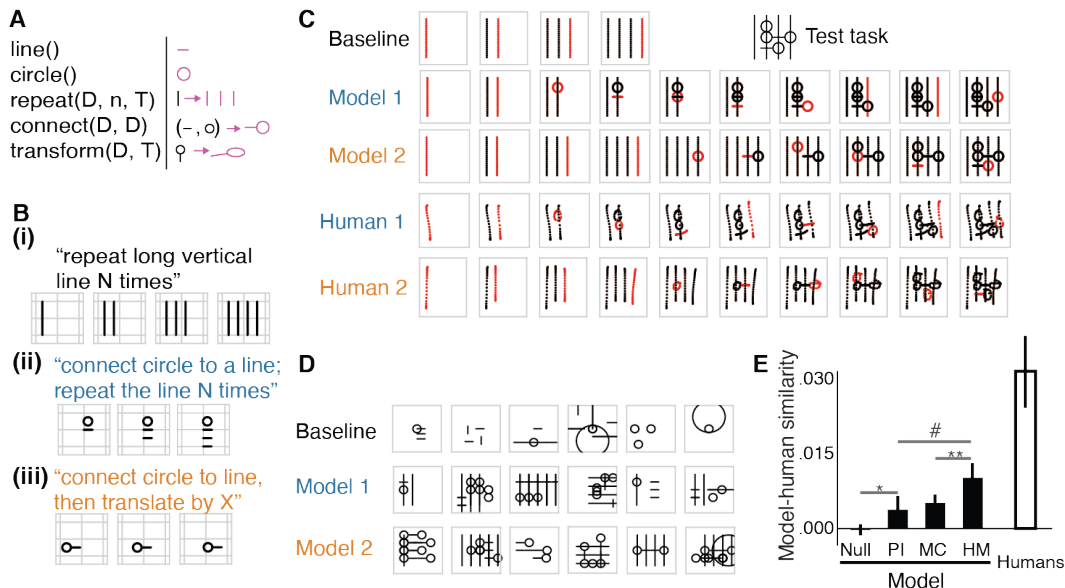

Figure 6: Modeling results. **(A)** Example starting primitives (left) and drawings (right). *D, n* and *T* are variables representing drawings, natural number, and transformation (see Table 1). **(B)** Example subroutines learned by both HM1 and HM2 (Bi), HM1 only (Bii), and HM2 only (Biii). **(C)** On an example test task, solutions by models (top) and example humans (bottom). **(D)** Example "dreams," or prior samples at Baseline, and after training on Tasks 1 (HM1) or 2 (HM2). **(E)** Comparing human and model behavior on test tasks. $Similarity = mean(D(H_1, M_2), D(H_2, M_1)) - mean(D(H_1, M_1), D(H_2, M_2))$, where distance $D(H_i, M_j) = \frac{1}{N_h}\frac{1}{N_s}\sum_{h \in H_i}\sum_s d(h, M_j, I_s)$ is distance to model averaged over humans ($h$) and test stimuli ($s$), for the group of humans trained on Task set $i$ and the model trained on set $j$. \*, \*\*, p<.05, .005; #, p=.06, paired t-test. Null model p<.05 vs. other models.

**Comparison with humans** We quantified the similarity between humans and models, finding that Human group 1 was better fit by HM1 than by HM2, and Human group 2 was better fit by HM2 than by HM1, indicating that learning altered the structure of behavior for both humans and models in overlapping ways (Figure 6E). Note that HM1 and HM2 used the same efficiency cost parameters (averaged over all subjects), so this "alignment" with humans is due to learned program structure. We also compared HM1 and HM2 to alternative "lesioned" models (see Methods and Table 2). HM performed better than Program Induction alone (PI), consistent with significant contribution of motor efficiency biases. HM also performed better than a model with learned motor biases but without program induction (MC), indicating an important role of abstract program-like structure in learning. As an upper bound, we assessed the similarity of humans to other humans in the same Training group (Figure 6E). Not surprisingly, compared to this upper bound, the HM model did not capture all

complexities of human behavior. This is partly due to human drawing being influenced by a variety of biases which we did not attempt to model (see Discussion). Taken together, these results argue that combining program induction and efficiency costs, with training on a small dataset and without access to human learning data, captures diagnostic features of human generalization.

# 4   Discussion

We show that humans spontaneously learn new abstract program-like structure from brief training on drawing tasks, and with no explicit supervisory signals for such structure. To understand this learning computationally, we built a generative model for drawing combined with a learning algorithm formalizing the principles of abstraction, compositionality, and motor efficiency. Trained on the same data as humans, this model learned a new set of abstract drawing subroutines by recombining a small set of simple drawing primitives. These learned subroutines support generalization behavior that resembles important aspects of human drawings in this task. Our results suggest that principles of abstraction and compositionality are central to explaining how humans learn generalizable program-like structure in drawing.

We formalize learning as acquisition of parsimonious internal models that explain shared structure underlying multiple learning experiences. The idea of knowledge as efficient abstraction has parallels in philosophy (e.g. *Occam's razor*), psychology [35]), and cognitive modeling (e.g. hierarchical Bayesian accounts of *learning-to-learn* [36, 24, 7] ). Our computational approach to learning builds on this work, by representing concepts as generative programs, and inductive biases as priors over programs [37, 38, 26, 27, 24].

One limitation of our study is that stimuli were relatively simple and "clean." This was by design, as we focus less on real-world writing or drawing skill (e.g., [24]) but on abstraction in rapid learning. Our approach can in principle generalize to learning more complex concepts, similar to human learning at a longer timescale, by providing the model with similarly generic starting primitives but a broader and more naturalistic training curriculum.

We also did not attempt to capture the full complexity of drawing, which likely contributes to the diversity of behavior across subjects. Future work may attempt to model this diversity as differences in starting priors (e.g., related to perception, motor skill, art and writing experience, and others).

Recent deep-learning-based models have had success modeling drawing and handwriting on more complex images than considered here. However, in contrast to our model, they need significantly larger training data-sets, in some cases supervising on human motor behavior [39, 40], and are usually not systematically compared to how humans learn new inductive biases [39, 41, 42, 40, 43]. The speed of human learning in our task, paralleled by the model, highlights the importance of learning rich and flexible structured representations for generalization, either explicitly [24, 44] or implicitly, as in differentiable neural computers [45] and others [46]. Similar to children acquiring sophisticated knowledge in a manner bootstrapped by "core" systems of knowledge [47, 48, 49], our study supports the view that the rapid learning of structured representations can result from appropriate starting primitives coupled to learning algorithms guided by abstraction and compositionality.

From an engineering standpoint, this work suggests and supports several directions in program induction. First, while prior work on modeling handwriting has shown the importance of compositionality [24], our work extends that to emphasize the importance of combining compositionality with higher-order abstraction in the form of library learning via symbolic compression. Second, by directly comparing the DreamCoder learning algorithm [23, 33] to human behavior, our work provides compelling evidence that this algorithm's integration of symbolic library learning with neural-network-guided search control is a promising approach for modeling some forms of human learning. Third, we found that incorporating an efficiency constraint led to more human-like performance. We suspect this last finding points toward the more general insight that program induction for planning problems may be improved by adding inductive biases so that programs are not only short (due to the minimum description length program induction prior) but also efficient when embodied as actions.

# 5 Broader Impact

We envision a number of scientific, societal and engineering benefits that may emerge from this study. First, this work may benefit the treatment, diagnosis, and prevention of cognitive disorders, such as disorders involving planning, reasoning, and learning. The methodology of our task may be particularly relevant for disorders that cause striking impairment in drawing behavior (for example, dementia, traumatic brain injury, and stroke). A computational understanding of cognitive impairment may lead to more accurate, quantitative diagnostic tools (by categorizing disorders based on cognitive computations) and to more efficient targeted treatment (by targeting of specific impairments).

Second, computational understanding of how humans think is insightful from a basic science perspective, because it advances our understanding of nature and the human condition. In addition to the current study of human adults, we are studying this task in children, and in non-human primates in a neurophysiological setting, with the goal of also studying this task at a neural level. One long-term goal of this multi-species investigation is to develop an evolutionary, developmental, and mechanistic understanding of how learning of complex structure can emerge from simple components and computational principles.

Third, from an engineering standpoint, this work may lead to AI that is more easily integrated into, and more beneficial to society. The ability to learn new human-like inductive biases from a small number of examples may facilitate human-computer interaction, particularly within programming-by-examples technologies [50]. Engineering outcomes of this research may also contribute to tools that benefit education. The link to learning drawing and art is obvious, but there may also exist links to topics that involve structured symbolic reasoning, such as math, science, or music. For instance, modeling a given student's learning trajectory may reveal what she knows and what strategies she uses to learn, which may suggest ways to either tailor her future learning, or to remedy current difficulties.

In principle, work along this line may potentially be used to create "fake" artifacts meant to pass as human. The most obvious kinds of fake artifacts are those related to drawing, but this extends to other kinds of art and media, including internet bots that impersonate humans by generating tweets from examples. One potential implication is that fakes will be used to confuse and manipulate society. Ways to address this should fall under strategies and considerations already being developed to understand the impact of increasingly human-like AI on society. A second implication is that mass-produced AI artifacts could lower the quality of creative content in the world, and compete with high-quality human-made creations. We think of this possibility as an ethical gray area, and note that it is an extension of the apparently already-occurring trend towards larger amounts of mass-produced media in society.

**Acknowledgments**   We thank Nathalie Fernandez, Brenden Lake, Max Siegel and João Loula and the Tenenbaum lab for helpful feedback. Kevin Ellis was supported by a NSF GFRP. Work supported by the NSF-funded Center for Brains, Minds, and Machines. The authors have no competing interests to disclose.

## Footnotes

[1]Types: $D$ is a drawing (set of coordinates that define a figure); $N$, $\theta$, $d$, $s$, $o$ are discretized parameters drawn from a multinomial distribution.

[2]ianli.github.io/raphael-sketchpad/

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
