[Supplementary Material]

# Supplement to: Learning abstract structure for drawing by efficient motor program induction

**Lucas Y. Tian**[1,2,3], **Kevin Ellis**[1,2], **Marta Kryven**[1,2], and **Joshua B. Tenenbaum**[1,2]

[1]Brain and Cognitive Sciences, MIT
[2]Center for Brains, Minds and Machines
[3]Laboratory of Neural Systems, Rockefeller University
{lyt,ellisk,mkryven,jbt}@mit.edu

## 1 Program synthesis algorithm

The algorithmic engine behind our program synthesis method follows the approach introduced in EC$^2$ [1], and is based on the open source implementation of EC$^2$'s successor, DreamCoder [2]. Our program induction model takes as input a corpus of black-and-white raster **training images**, and seeks to synthesize a graphics program that generates each of them. The model estimates a prior over programs for training images, to be deployed on held-out **test images**. Following [2] we now derive this algorithm starting from a Bayesian viewpoint. With this probabilistic formalism in hand we will then briefly outline the 3-step algorithm which performs inference in this model, but interested readers should consult [2, 1] for a complete algorithmic exposition.

**Notation.** We write $I$ to mean an image, and $\rho$ to mean a graphics program. Programs are represented in typed lambda calculus. Initially the graphics programming language contains the **primitives** outlined in Table 1 of the main text. Primitives are expressions in typed lambda calculus.

We write $[\![\rho]\!]$ to mean the image output by program $\rho$. We write $L$ to mean a **library** of primitives; At the initial state of learning $L$ contains the primitives in Table 1 of the main text. The library $L$ acts as a prior over the space of programs, written $P(\cdot|L)$ and defined formally in [2]. Intuitively, this prior prefers programs which may be expressed compactly using the primitives in $L$. We write $P(L)$ to mean the prior probability of the library $L$, and this prior prefers libraries which overall contain less code (smaller lambda calculus expressions).

From a Bayesian point of view our aim is to estimate the prior maximizing the joint probability, which we will notate $J$:

$$P(L|\{I_n\}_{n=1}^{N}) \propto P(L, \{I_n\}_{n=1}^{N}) = J(L) = P(L) \prod_{n=1}^{N} \sum_{\rho} P(I_n|\rho)P(\rho|L) \qquad (1)$$

where $P(I|\rho) = \mathbb{1}\left[I = [\![\rho]\!]\right]$. Evaluating this objective is intractable because it requires marginalizing over the infinite set of all programs. We define the following intuitive lower bound, written $\mathscr{L}$, on this objective function:

$$J(L) \geq \mathscr{L}(L, \{\mathcal{B}_{I_n}\}_{n=1}^{N}) = P(L) \prod_{1 \leq n \leq N} \sum_{\rho \in \mathcal{B}_{I_n}} P(I_n|\rho)P(\rho|L) \qquad (2)$$

where the bound $\mathscr{L}$ is expressed in terms of a collection of sets of programs, $\{\mathcal{B}_n\}_{n=1}^{N}$, called **beams**:
**Definition.** A **beam** for image $I$ is a finite set of programs where, for any $\rho \in \mathcal{B}_I$, we have $P(I|\rho) > 0$. In other words, every program in the beam for image $I$ correctly draws $I$.

Making the beams finite ensures that calculation of $\mathscr{L}$ is tractable. In our experiments we bounded the size of the beams to 5.

We alternate maximization of $\mathscr{L}$ with respect to the beams and the library. In reference to figure 2 of the main text, these alternate maximization steps are called the **Explore** and **Compress** steps.

**Explore: Maxing $\mathscr{L}$ w.r.t. the beams.** Here $L$ is fixed and we want to find new programs to add to the beams so that $\mathscr{L}$ increases the most. $\mathscr{L}$ most increases by finding programs where $P[I|\rho]P[\rho|L]$ is largest.

**Compress: Maxing $\mathscr{L}$ w.r.t. the library.** Here $\{\mathcal{B}_{I_n}\}_{n=1}^N$ is held fixed, and so we can evaluate $\mathscr{L}$. Now the problem is that of searching the discrete space of libraries and finding one maximizing $\mathscr{L}$.

Searching for programs is hard because of the large combinatorial search space. We ease this difficulty by training a neural recognition model, $Q(\cdot|\cdot)$, during the Compile step: $Q$ is trained to approximate the posterior over programs, $Q(\rho|I) \approx P(\rho|I, L)$, thus amortizing the cost of finding programs with high posterior probability.

**Compile: learning to tractably maximize $\mathscr{L}$ w.r.t. the beams.** Here we train $Q(\rho|I)$ to assign high probability to programs $\rho$ where $P(I|p)P(p|L)$ is large, because including those programs in the beams will most increase $\mathscr{L}$. We train $Q$ both on programs found during the Explore step and on samples from the current library, i.e. $P(\cdot|L)$. Assuming that $Q$ successfully converges to the true posterior estimates, then incorporating these top programs as measured by $Q$ into the beams will maximally increase $\mathscr{L}$.

## 1.1 Algorithmic details

Having introduced the probabilistic framing of our problem, and the 3-step inference procedure, we now briefly outline the algorithmic implementation of the explore/compress/compile steps. A full overview is contained in [2].

### 1.1.1 Explore

During the Explore step we enumerate programs in decreasing order under $Q(\cdot|I_n)$ for each image $I_n$, and keep the top 5 within the beam $\mathcal{B}_{I_n}$ as measured by the posterior $P(\rho|I_n, L)$. This enumeration is tractable given our parameterization of $Q$, which can be expressed similarly to a PCFG over programs; i.e., the neural network outputs a distribution over programs parameterized by the weights of a probabilistic grammar, and we enumerate in decreasing order under that grammar. Thus, $Q$ outputs the transition probabilities of a bigram model over program syntax trees, which may be unrolled into a PCFG-like representation. Enumeration proceeds until a per-image timeout is reached; we used a timeout of one hour.

### 1.1.2 Compress

Here we seek to update the library by increasing the probability it assigns to programs in the beams, hence "compressing" those programs. Indeed the compress objective can be rewritten in terms of a sum of description lengths:

$$\arg\max_L \mathscr{L} = \arg\min_L \underbrace{-\log P(L)}_{\text{description length of library}} + \sum_{1 \leq n \leq N} \underbrace{-\log \sum_{\rho \in \mathcal{B}_{I_n}} P(I_n|\rho)P(\rho|L)}_{\text{description length of program generating image } I_n} \tag{3}$$

To heuristically minimize this description length we search locally through the space of libraries $L$ until the above objective fails to improve. Our search moves consist of incorporating new subexpressions obtained from automatically refactoring programs in the beams—intuitively, refactoring programs that we found explaining images so as to minimize the total size of the library plus the total size of those programs. See Supplemental Figures 1 and 2 for new subexpressions learned during training. This refactoring process combines version space algebra [3] with equivalence graphs [4], which are two approaches from the programming languages and program synthesis community; see [2] for details.

Figure 1: New functions learned after training on Task set 1. Each learned function is a new primitive that can be reused to build increasingly complex functions (indicated by arrows). The starting primitive functions (e.g., transform, repeat, ...) are described in the main text (Table 1), which "transmat" here equivalent to "affine" in Table 1. The parameters whose names begin with "scale", "angle", "dist", "rep" correspond to sampled values for $s$, $\theta$, $d$, $N$ in Table 1.

Figure 2: New functions learned after training on Task set 2. See legend for Figure 1.

### 1.1.3 Compile

Here we train a neural network to guide the search over programs, seeking to minimize its divergence from the true posteriors over programs. Writing $\phi$ for the parameters of $Q$, we aim to maximize

$$\arg\max_{\phi} \mathrm{E}\left[Q_{\phi}\left(\left(\arg\max_{\rho} P(\rho|L, I)\right) \mid I\right)\right] \tag{4}$$

where the expectation is taken over images $I$. Taking this expectation over the empirical distribution of images trains the network on programs found during the Explore step; taking the expectation over samples, or "dreams," from the learned prior $L$ is critical for sample efficiency: just like humans our model learns from at most a few dozen images, which is too little training data for a high-capacity neural network. But as we learn our prior, we can then draw unlimited dreams to train the neural network.

## 1.2 Generalizing to test images

Prior to evaluation on test images we iterate this learning procedure for 20 cycles (of searching for task solutions, updating the library, and training the neural network). The end state of learning is not just a program for each training image but, critically, also a learned inductive bias $L$ and a learned inference/synthesis strategy $Q$. When comparing with human data we infer a program for test images by enumerating programs in decreasing order under $Q(\cdot|I)$ and then rescoring under $P_{\text{test}}(I|\rho)P(\rho|L)$, where the likelihood $P_{\text{test}}(I|\rho) \propto \exp{(-|I - [\![\rho]\!]|_2)}$ is a relaxed version of the 0/1 likelihood during training to allow partial credit when the model cannot fully explain a test image.

# 2 Analysis of behavior

## 2.1 Converting a motor trajectory into a sequence of discrete strokes

Motor trajectories [raw data in the form of coordinates and corresponding times $(x, y, t)$] were segmented into discrete "strokes". Each stroke was a sequence of coordinates during which the finger was continuously touching the screen; i.e., strokes were separated by no-touch gaps. Subjects naturally tended to lift their finger between each discrete segment (i.e., line or circle) in the drawing.

Each stroke was summarized in a stroke-level feature vector $\psi_{stroke} = (category, startLocation, center, row, column)$. $category$ is the category of action represented by the stroke, either a "vertical line" (LL), "horizontal line" (Lh), or "circle" (C). $startLocation$ is the $(x, y)$ position of the stroke onset. $row$ and $column$ were discretized values defined by "snapping" a stroke onto its position in a $3 \times N$ (row $\times$ column) grid, where $N$ is the number of vertical lines in the "grating" for a given stimulus.

An entire trajectory $t$ was therefore defined by the ordered list of strokes: $(\psi_{stroke}^1, \psi_{stroke}^2, ...)$.

To calculate frequencies of horizontal/vertical transitions in a given trajectory, stroke transitions were classified as either horizontal (same row, different column), vertical (any row, same column), or undefined (different row, different column).

## 2.2 Computing distance between two trajectories.

$dist_{traj}(t, t')$ computes a scalar distance between trajectories $t$ and $t'$, based on a string-edit distance. Each trajectory was converted to four different strings (each with length equal to number of strokes), each representing a sequence of a stroke-level feature: (1) action category (i.e., circle, horizontal line, vertical line) (2) column location (3) row location; and (4) conjunction of the first three features. These features are described in Suppl. Sect 2.1. Each feature contributed a scalar distance using the Damerau–Levenshtein string-edit distance; these distances were averaged to compute $dist_{traj}(t, t')$.

## 2.3 Motor Cost model: extracting motor cost features from motor trajectories

We define a *feature extractor* $\phi(\cdot)$ that maps a trajectory $t$ to a trajectory-level real-valued feature vector $\phi(t)$ with four elements based *a priori* on previously described drawing biases [5, 6]: $start$, $distance$, $direction$, and one meant to capture biases reflecting learning in this task $verticality$.

$start$ is position of first touch, represented as distance from the top-left corner. $distance$ is the summed distance traveled, measured as the path between stroke centers, $direction$ is the summed distance travelled, but projected onto the diagonal from top-left to bottom-right, a direction chosen because it is a common bias in sketching and writing. $verticality$ is the cumulative distance moved projected onto the y-axis subtracting distance projected onto the x-axis, and was included to allow the Motor Cost models trained on separate groups (MC1 and MC2) to capture learned directional biases.

## 2.4 Extending motor cost features to quantify program-like structure in behavior

In order to quantify variation in behavior across subjects, we fit Motor Cost model parameters to each subject, but with the model extended with two parameters in addition to the four described above, $chunking$ and $skewers$. $chunking$ was the count of the number of transitions between strokes of identical categories (e.g., circle -> circle), which reflects a bias to group similar objects together. $skewers$ quantifies whether transitions away from "long vertical line" tend to be vertical (reflecting

a bias to draw skewers) or horizontal (reflecting a bias to draw gratings); this was implemented similarly to *verticality*. Each subject was therefore described by a single 6-dimensional feature vector $\theta$.

Subjects exhibited program-like structure as described in Figure 4 in the main text. Assignment of subjects to different strategies was supported both by visual inspection of their behavior and clustering of subject-level feature vectors $\theta$, from fitting this extended Motor Cost model (Supplemental Figure 3). We found the outcomes of these two methods to be largely in agreement. We note that subjects did tend to form a continuum between these four strategies, and so we attempted to assign each subject to their dominant strategy.

Figure 3: Quantitation of diversity of program-like strategies. Left scatter plots: datapoints represent values of Motor Cost model features for individual subjects along three dimensions: "skewers", "vertical", and "typechunking" are Motor Cost model features. Top row: first, subjects were grouped into those drawing "skewers" and those drawing "gratings" (Step 1). Bottom row: second, the "gratings" group was split into three groups based on what they drew immediately after drawing gratings (Step 2). These two steps led to four mutually exclusive groups. Colored boxes allow for comparison between the scatterplots (left) and example motor sequences (right).

## 3 Code and datasets

Code and datasets can be downloaded from: https://www.dropbox.com/s/ehy6w0j1sippt95/code.zip?dl=0