[Reviews · NeurIPS 2020]

Review 1

Summary and Contributions: This paper compares a computational model of learning drawing programs to a survey on human subjects. It shows similarities between human subjects and a learned computational models in terms of the abstract structure of the learned programs.

Strengths: The paper describes a thorough experiment on human subjects. The paper analyses the results using a state of the art computational model of program induction. The results are very interesting, shed light on human intelligence and can serve as inspiration for artificial intelligence, and drive further research on models for program inductions and learning abstract structures and concepts.

Weaknesses: There's no direct conclusion on how to improve program induction models given the presented results.

Correctness: As far as I can tell everything is correct.

Clarity: The paper is very well written, see some minor comments below.

Relation to Prior Work: As far as I can tell this is clearly presented although I am not too familiar with the relevant literature.

Reproducibility: Yes

Additional Feedback: Some aspects of the computational model are not clear. 1. Lines 94 - 106 describe a way to bias the model towards efficient motor control without supervision of the motor data, why isn't this use in PI models in table 2 ("Admissible traj., all equal probability")? 2. Most of the results are using the hybrid models which are trained using the motor data. This makes the training set different than for the human subjects which only see the images. How is this comparable? I think this should be discussed. Some more comments: 1. How consistent are models trained on the same data? It would be good to see a plot like 3c and 3d on models trained with different random seeds. 2. It would be useful to see the updated libraries after training. 3. Section 3,3 describes the results in figure 6 mentioning the models (HM1 and HM2) but the figure mentions PI1 and PI2. Not sure which are the correct ones. Line 182 refers to table 2 instead of table 1? Update: I have read the authors' feedback and happy to keep my score.


Review 2

Summary and Contributions: This is a cognitive modeling paper, combining results of a human subjects experiment on copying simple drawings with a Bayesian model of program induction. Properties of generalization were measured by using two different training sets and measuring properties on the same test set, for both people and the model. Ablated versions of the model were used as well, to argue that the results were due to the claimed factors.

Strengths: The theoretical description was clear and the empirical evaluation was performed well. Since learning is central to NeurIPS, and there is a neurosymbolic component to the model, it should definitely be of interest to this community.

Weaknesses: There are two weaknesses. The first is that this is an incremental step along a path trod by a number of other papers in PNAS, Science, etc. It is not clear how much is new here, compared to those prior efforts, including the two DreamCoder papers. That should be clarified in a revised version. The second weakness is that the method of randomly generating programs until you find something that works ("programs are sampled from a generative model...") is unlikely to scale to more complex drawing tasks. This is the same limitation that prior work in this genre has. By contrast, models like SOAR, which accumulate procedural knowledge via chunking, would become more skilled over time, following the power law of learning. Would the approach described here do that? It might, if the update of the library were handled properly, since the power law of learning falls out from any incremental knowledge accumulation method. That would be an interesting test of this kind of model.

Correctness: To the extent that can be determined from a conference paper, the claims, method, and methodology are correct.

Clarity: The writing is very clear.

Relation to Prior Work: As noted above, how this differs from prior work in Tennenbaum's group has not been adequately delineated.

Reproducibility: Yes

Additional Feedback: I think this is a strong paper, in a line of research that is interesting and should be of interest to the NeurIPS community. Making it clearer how this work constitutes an advance over the prior work in this line is the major point I would like to see addressed in the revision.


Review 3

Summary and Contributions: The goal is to study how humans learn to perform a novel task via abstraction and compositionality of simpler routines. The experimental manipulation is to use two different types of training sets that are designed to lead participants to learn different inductive biases. The same test set is used for both groups. Thus, systematic differences in how subjects complete the task across groups may be attributed to differences in how they learned the task, due to the assignment of the training set. A program induction algorithm for learning the task demonstrates the same types of biases as seen in the humans. Also, the outputs of the program learned on training set 1 are more similar to the human drawing trajectories for group 1 than for group 2, and vice-versa for outputs of program learned from training set 2.

Strengths: The experimental design controls for confounding factors by randomization. The stimuli design is quite ingenious in allowing two different types of drawing strategies to be tested on a common set. The program induction uses state-of-the-art probabilistic inference methods to learn how to draw, and qualitatively replicates important features of human performance. A second experiment with a rotated test set adds extra confirmation that humans learned the task as a composition of simpler tasks. The data displayed is remarkably clean and shows high compliance of the human volunteers to the intent of the task. This study is very interesting for the researchers interested in understanding human learning, and encourages the use of similar methodology for future studies in human task learning.

Weaknesses: As the authors note, the AI has much less agreement to human behavior than intersubject agreement within the same group. But due to the novelty of the work, such a performance gap is understandable. The authors used very strong prior knowledge (e.g. pre-defined primitives) which while intuitively plausible, may limit the generality of the AI.

Correctness: The algorithmic and the experimental methodology appear to be correct.

Clarity: On the whole, the paper is quite readable. One issue that was not immediately clear is the terminology of image, stroke, trajectory. The meaning of the notation in the equations (4) "t draws I" is not clear. I had to refer to the supplement to understand that the trajectory did not refer to the entire image but rather to a single stroke.

Relation to Prior Work: Previous literature from two fields, (1) psychological studies of drawing and (2) AI program induction and drawing are cited. This study is novel in comparing human data to AI behavior.

Reproducibility: Yes

Additional Feedback: The paragraph "Reweighting trajectories by motor cost" could probably be made more clear. It was not clear if all the details needed to reproduce the implementation of the program induction are given. But these are relatively minor issues and would not improve the already high score that I am giving this paper.

[Author Response · NeurIPS 2020]

R1 raises a concern about the engineering relevance of our work, saying there is "no direct conclusion on how to improve program induction models". While our focus is cognitive modeling, we do also hope to guide research in program induction. This can come in at least two ways. (1) our results show that an approach integrating library learning (via symbolic compression) and learned search control (via a neural network) is a promising way to describe human learning. This motivates further exploring these approaches in program induction in AI. (2) We discovered that a learned syntactic simplicity bias over the space of programs (library learning) does not suffice for inferring human-like motor plans. Instead, a hybrid model combining this simplicity bias with a bias toward efficient motor plans was needed. We suspect this finding points toward a more general insight, that program induction for planning problems would be improved by adding inductive biases in favor of short *and* efficient programs. Here efficiency could be AI-specific (e.g computability, ergonomics), or could be based on human motor constraints, to generate human-like behavior. We will revise our paper to highlight these points.

R1 asks why efficiency biases (described in Lines 94 - 106) were not applied to the PI model. We did not bias the PI model because it acts as an "efficiency lesioned" comparison to the Hybrid model (which we did bias).

R1 raises a concern about how best to compare human and model behavior, given that part of the training of the Hybrid model was supervised by motor data, while humans were not. The supervision by motor data was used to infer efficiency constraints that humans naturally bring to the task. From a human's perspective, these biases are already present, so there is no need to further "train" on these biases. Comparing different models, trained with and without efficiency biases, with humans thus highlights the importance of capturing this human prior knowledge in the model.

R1 asks about consistency of models trained with different random seeds. We have observed the results of learning to be relatively stable over random seeds. But we thank the reviewer for raising this important point. We can prepare supplemental figures showing results from several other random seeds.

R1 asks for illustrations of libraries post-training. Figure 6B,D show subsets of the updated libraries after training, as well as sampled programs from those libraries, but we agree that this core aspect of our model deserves further illustration. We can prepare supplemental figures showing a wider range of learned library routines.

R1 asks which models are shown in Figure 6. All of the model results are for the Hybrid model, except when noted otherwise: "Baseline" in Panels C and D; "Null", "PI" and "MC" in Panel E. We thank the reviewer and can update the figure legend accordingly. We also thank the reviewer for bringing to our attention the error in line 182.

R2 is concerned that this work is incremental relative to DreamCoder/EC2 and Lake et al. 2015. While we build on code of the former and ideas of the latter, there are qualitative differences. DreamCoder/EC2 neither compare model with human behavior nor learn efficient plans, and these issues are intertwined: as mentioned earlier, we discovered that efficiency biases are needed in tandem with simplicity biases to best account for human behavior, which may have repercussions both in our computational understanding of this behavior, and in how we design program inductors for planning problems. Learning in Lake et al did not implement library learning via symbolic compression (therefore lacking higher-order abstractions in drawing/handwriting) and did not use learned search control (via a neural network). Our results highlight the importance of these two features for modeling human learning.

R2 raises a possible confusion about our inference algorithm. To be clear, the model does not search randomly for programs until finding one that works, but instead performs a neurally-guided systematic search. The quotation "programs are sampled from a generative model..." refers to how training data are generated for this neural guidance. Our library learning is similar to "chunking", with chunks discovered via a refactoring step, and does indeed become "more skilled over time" as R2 asks: at the start of learning, none of the training images can be drawn (0/72 combining both training sets) but at the end state, almost all of them can be (68/72).

R4 points out that this work is just a first step in capturing rapid learning of motor plan generalizations, as highlighted in Figure 6E by the gap between (model vs human) and (human vs human) agreement. Indeed, we only attempt to capture a subset of the myriad experiences that influence learned structure (and via only a few dozen training stimuli). We hope that future work helps close this modeling gap.

As R4 notices, we used pre-defined starting primitives (circles, lines, repetitions, etc.), which are relatively simple yet plausibly accessible to humans even before they begin the study. Our approach can in principle generalize; e.g., to model learning at a longer timescale (e.g., developmental), we could start with a spartan yet highly generic basis, and learn to draw from a broader training curriculum. This would be exciting future work, although with less relevance for the behavioral experiment designed here for studying rapid single-session learning of higher-order structure. We thank R4 for this possible future direction.

We thank R4 for suggesting improving readability of the section "Reweighting motor trajectories by motor cost". We will update this section with the suggested modifications: (1) clarifying that "$t$ draws $I$" means the trajectory $t$ (a sequence of strokes) produces the image $I$; (2) moving relevant parts of Suppl. 2.2 to this methods section.

[Meta-Review · NeurIPS 2020]

Three knowledgeable referees support accept and I accept. We encourage and expect the authors to incorporate the reviewers' suggestions for improving the paper.